# Quality of Life and Physical Activity of Persons with Spinal Cord Injury

**DOI:** 10.3390/ijerph18179148

**Published:** 2021-08-30

**Authors:** Tjasa Filipcic, Vedrana Sember, Maja Pajek, Janez Jerman

**Affiliations:** 1Faculty of Education, University of Ljubljana, Kardeljeva Ploščad 16, 1000 Ljubljana, Slovenia; tjasa.filipcic@pef.uni-lj.si (T.F.); janez.jerman@pef.uni-lj.si (J.J.); 2Faculty of Sport, University of Ljubljana, 1000 Ljubljana, Slovenia; vedrana.sember@fsp.uni-lj.si

**Keywords:** spinal cord injury, sports activity, quality of life

## Abstract

The higher quality of life of people with spinal cord injury is closely related with their reintegration into the social environment. Social reintegration is a demanding and complex process, requiring individuals to become active again and acquire age-, gender-, and culture-appropriate roles and social status. It also involves independence and productive behavior as part of multiple interpersonal relationships with family, friends, and others. In order to establish whether individuals with spinal cord injury who are physically active subjectively rate their quality of life to be higher compared to those who are not, sixty-two respondents from Slovenia with spinal cord injury were interviewed. Thirty-one of them were physically active, and 31 were not. The level of injury of the responders was from Th6–Th12. The participants gave the highest assessments to their interpersonal relationships, and the lowest to their satisfaction with material prosperity. Data comparison showed that subjective estimates in all areas of quality of life are higher in respondents who were involved in physical activity after their injury. The results may encourage persons with spinal cord injury to participate more often in sports programs, and also encourage others to do so.

## 1. Introduction

Quality of life (QL) is considered as an objective (achievements) or subjective (expectations) entity and refers to a broad category of phenomena that includes individual satisfaction with life domains. Since individuals’ achievements and expectations change over time, both objective and subjective QL are dynamic [1]. QL may change when an individual suffers a traumatic injury, such as a spinal cord injury (SCI). It results in chronic motor and sensory impairments that lead to lifelong disability [2] and affect QL.

Individuals with SCI report poorer QL than those without disabilities, as reported by Dijkers [1]. Post and Van Leeuwen [3] reported that differences in ratings of QL between people without and with SCI are quite large (effect size (ES) = 0.77). Among researchers and clinicians, the focus has shifted to identify predictors of QL and developing interventions to improve QL in individuals with SCI. One such intervention that has been shown to improve QL is physical activity (PA). Sweet, Martin Ginis, and Tomasone [4] state that regular physical activity is an important factor that can have a positive impact on several domains of quality of life. WHO, in 2020 [5], introduced guidelines for PA for people with disabilities. It is recommended to perform at least 150–300 min of moderate-intensity aerobic exercise PA or 75–150 min of vigorous-intensity aerobic exercise PA per week to achieve health benefits. Martin Ginis et al. [6] recommended moderate to vigorous intensity aerobic exercise of at least 20 min twice per week and strength training at least twice per week to observe changes in QL outcomes. Some studies, for example [4,7,8], reported large, statistically positive associations between PA and QL in individuals with SCI, while others showed smaller or even nonsignificant associations [9,10]. Ginis et al. [11] wrote that these results likely reflect methodological differences in QL variables that varied across studies, and researchers used different measures for PA such as frequency, intensity, duration, and type of PA. In the same study [11], researchers meta-analyzed 21 studies that examined the relationship between PA and QL. They found statistically significant small to moderate effects on the relationships between PA and QL, PA and depressive symptoms, and PA and life satisfaction. Studies with experimental and quasi-experimental designs reported larger effects for QL and life satisfaction than studies with nonexperimental designs. Of the 21 studies included in the analysis, only three studies contained a homogeneous sample (paraplegia) with a rather small number of participants [12,13,14]. All other studies included in the meta-analysis by Ginis et al. used mixed samples (paraplegics, quadriplegics with different level of injury, or individuals with other disabilities).

Tomasone et al. [15] conducted another meta-analysis of 33 studies examining QL (objective and subjective) and PA in individuals with SCI (paraplegics and quadriplegics). The results suggest that PA is significantly associated with an increase in objective and subjective QL, while relatively few studies show a negative or nonsignificant association relationship. Of the 33 studies, only two studies [13,16] included a homogeneous sample with a small number of paraplegics (15 in the first and three in the second study), while all other studies included paraplegics and quadriplegics. Based on the literature, it is reasonable to design such a study that would include individuals with similar level of injury. Tomasone et al. also formulated some other implications for further research. They suggested that more research should be conducted to examine the subjective QL and its relationship to PA, and that more attention should be paid to the social domain of QL. 

To follow the suggestions of Martin Ginis, and Tomasone et al., the aim of our study was to investigate the subjective QL, considering also the social domain. We developed an instrument to measure QL among people with SCI. The focus was on whether there are statistically significant differences in subjectively perceived QL between physically active and inactive individuals with complete thoracic SCIs (Th 6–Th 12). Areas of QL were: material prosperity, physical and emotional wellbeing, personal development, self-determination, interpersonal relationships, social integration, and rights.

## 2. Materials and Methods

### 2.1. Participants

Sixty-two subjects with complete thoracis SCI participated in the study. According to their level of physical activity, they were divided into two groups: (i) physically active and (ii) physically inactive. They formed the probability sample (random). The basic criteria used to select participants with SCI included traumatically acquired complete SCI below the Th 6 level–Th 12 level, with time since the injury occurred not less than two years. The age of the participants at the time of measurement was between 25 and 68 years (48.3 ± 9.9 years). The sample of all persons with SCI consisted of 47 males (75.8%; 48.3 ± 10.0 years) and 15 females (24.2%; 48.0 ± 9.7 years).

The physically active group included 27 males (87.1%; 43.3 ± 9.3 years) and 4 females (12.9%; 46.0 ± 3.6 years), and the physically inactive group included 20 males (64.5%; 55.2 ± 6.3 years) and 11 females (35.5%; 48.7 ± 11.3 years). The groups differed in age at the time of examination (t = 4.17, *p* = 0.001) and age at the time of injury (t = 2.35, *p* = 0.022), but not in time elapsed since injury (t = 0.86, *p* = 0.39), gender (χ^2^ = 3.16; *p* = 0.75), marital status (χ^2^ = 3.75; *p* = 0.29) (most respondents were married), education (χ^2^ = 6.03; *p* = 0.19) (most respondents had completed secondary school gymnasium), and employment (χ^2^ = 7.38; *p* = 12) (most respondents were retired due to disability).

### 2.2. Data Collection

Aims of the research were initially presented to the SCI Association (SCIA) across the Republic of Slovenia. Persons with SCI were invited to fulfil the questionnaire (subjective QL among persons with SCI, Ljubljana, Slovenia). Their collaboration was voluntary, the questionnaire was anonymous, and written informed consent was obtained from all participants included in the study. Each individual had sufficient time to complete the questionnaire and was given addition explanation if needed. The research was performed before the COVID-19 pandemic; therefore, face to face communication was possible.

A total of 62 persons responded and completed the questionnaire. The study was conducted according to the Declaration of Helsinki and approved by the Human Ethics Committee of the Faculty of Education, University of Ljubljana (34/12.3.2019).

To meet the requirements of quantitative research, the survey technique using a closed-ended questionnaire was used. The questionnaire contained statements that respondents answered using a five-point Likert scale. The instrument contained eight areas of QL that were presented by Schalock and colleagues [17]. 

The instrument was first tested with a pilot study. The objectivity of the survey was enhanced by clear instructions, and the objectivity of the evaluation of the responses was ensured by the use of closed questions and clearly defined categories in the rating scales. The reliability of the questionnaire was considered using intraclass coefficient (ICC) and Cronbach’s alpha. ICC was calculated by a two-way random analysis and showed good value: 0.87. Cronbach’s alpha found to be a very high value: 0.95. Content validity was independently assessed by five experts in the field of QL of persons with SCI. Content validity ratio (CVR) was found to be excellent: 0.99 (unpublished data, 2020).

### 2.3. Procedure

For the purposes of the present research, we used the following variables: (i) physical activity: respondents were asked to define themselves as physically active (when they reached at least 150 min of moderate intensity aerobic physical activity or at least 75 min of vigorous intensity aerobic physical activity throughout the week) or inactive (not engage in any physical activity that would reach criteria for active responders). We followed WHO guidelines 2020 [5]. Further on, we collected data regarding (ii) material prosperity (satisfaction with income, the ability to take care of oneself and one’s family, ability to work, obstacles—employment; (iii) physical wellbeing (health, obstacles—daily tasks, capacities—daily tasks, leisure activities); (iv) personal development; (v) emotional wellbeing; (vi) self-determination; (vii) interpersonal relationships; (viii) social inclusion; (ix) rights (please see Appendix A ).

### 2.4. Data Analysis

The data were analyzed using IBM SPSS Statistics for Windows, Version 22 (SPSS Inc., Chicago, IL, USA). Data were presented as mean values (mean) and standard deviations (SD). The basic descriptive statistics of the numerical variables (measures of central tendency and measures of dispersion) were calculated, and the normality of the distribution was checked by conducting the Shapiro–Wilk test. The Levene’s test was used to validate the homogeneity of variance. Differences were tested by applying the t-test for independent samples. To account for multiple testing, we adjusted *p* values of these associations with the use of the Benjamini–Hochberg false discovery rate procedure (FDR) [18]. The magnitude of difference was measured by using Cohen’s d and interpreted as follows: trivial: 0.0–0.2; small: 0.2–0.5; moderate: 0.5–0.8; large: ≥0.8.

## 3. Results

### 3.1. Population

Sixty-two volunteers (47 men and 15 female), (mean age 48.3) completed the questionnaire. The results are presented in the same order (Table 1, Table 2, Table 3, Table 4, Table 5, Table 6, Table 7 and Table 8) as the areas and dependent variables given in Appendix A.

### 3.2. Material Prosperity

Table 1 shows that all the average estimates of the variables that define material prosperity are higher for individuals who are physically active. The active group reported significantly higher values for the variables representing “satisfaction with income”, “ability to take care of oneself or one’s family”, and the “ability to work”, but values were insignificant in the case of the “obstacles for employment” variable. The effect size indexes are high for all variables except for the variable “obstacles for employment”.

### 3.3. Physical Wellbeing

Table 2 shows that all the average estimates of the variables that define physical wellbeing are higher for persons who are physically active. The difference is statistically significant in all variables except in the case of “the obstacles to daily tasks” variable. This also applies to the effect size of the indexes, namely, all the effect size indexes are high except for the variable “obstacle to daily tasks”.

### 3.4. Personal Development

Table 3 shows that all the average estimates of the variables relating to personal development are higher for individuals who are physically active. The difference is statistically significant in all variables, while the effect size indexes are the highest in the following variables: “productivity”, “overcoming obstacles”, and “personal competence”.

### 3.5. Emotional Wellbeing

Table 4 shows that all average estimates of the variables that define emotional wellbeing are higher for individuals who are physically active. The difference is statistically significant in all the variables except for “religiosity”. The same also applies to the effect size indexes.

### 3.6. Self-Determination

Table 5 shows that all the average estimates of the variables relating to self-determination are higher in individuals who are physically active. The difference is statistically significant for all variables, while the effect size indexes are the highest in “autonomy” and “personal control”.

### 3.7. Interpersonal Relationships

Table 6 shows that all average estimates of variables that refer to interpersonal relations are higher in individuals who are physically active. The difference is statistically significant for all variables. The index of the effect size is very high for the variable “leisure time with friends”, while other effect size indexes are moderate.

### 3.8. Social Inclusion

Table 7 shows that all average estimates of the variables defining social inclusion are higher for individuals who are physically active. However, the difference is statistically significant only in three variables, which is also shown in high or medium effect size.

### 3.9. Rights

Table 8 reveals that the average estimates of the variables that define rights are higher in individuals who are physically actives. The difference is statistically significant in all variables, which also applies to the effect size index in these variables; the only exception is the variable “timeliness of processes”.

## 4. Discussion

In the current study, the participants gave the highest assessments to their interpersonal relationships, and the lowest to their satisfaction with material prosperity. Data comparison showed that subjective estimates of satisfaction in all areas of QL are higher in respondents who were involved in PA.

The results of the analysis in the area of material prosperity are confirmed by research in the literature. Although employment of persons with SCI is generally low, it has been proven that greater opportunities for retaining employment, acquiring a new job, or starting training for a new profession are also related to participation in PA [19]. PA contributes to faster recovery of lost power and skills, less secondary conditions, or health problems after the injury, as well as to lower fatigue. These are important factors which increase employability [20]. Other positive effects of engagement in PA (including control of the secondary conditions, greater functional independence, faster recovery, acquisition and retention of lost strength, better mental status) also make it easier for a person with SCI to return to the workplace or contribute to acquiring new employment, because PA enhance general work capacity, ability, and performance [21]. Furthermore, if a person is satisfied with the payment they receive and the work they do, their inner satisfaction increases, along with the quality of their life [20,22,23,24,25].

The maintenance of health is one of the most important topics for persons with SCI. Such an injury is accompanied by complex secondary conditions that can have a very negative impact on the QL. Adapted PA can help to prevent or reduce major secondary conditions after injury and the need for prolonged hospitalization or re-hospitalization [26]. Restoring, acquiring, and maintaining lost strength and general physical fitness strengthens the immune system, reduces pain [27], and improves the cardiopulmonary abilities and the cardiovascular system [28,29]. In our survey, persons with SCI who were physically active also devoted more time to recreational pursuits. It is probably the resulting positive experiences, sense of comfort, and the relaxation of the body and spirit that contribute to greater internal motivation for more regular PA in the daily schedules of persons with SCI [30]. Obstacles that appear after SCI with regard to performing daily tasks are part of the reality of life for such individuals, and all the respondents experience them in a similar way. However, those who are physically active tend to do better, which means that they are less likely to be hindered in their daily tasks than those who are physically inactive. This can be linked to the aforementioned positive effects of engagement in PA.

PA has been seen to positively impact personal development of persons with SCI. In our study, it was observed that PA attributes to higher desire for achievements and ability to focus on learning. Active individuals are more satisfied with personal qualification such as cognitive, social, and motor abilities. They experience success and own productivity on a higher level, compared to those who are inactive. Kennedy and colleagues [31] observed increased general self-efficacy in active persons with SCI, while others [32] noted the higher self-evaluation of physical abilities, leading to greater emotional stability, psychological adjustment, and sense of independence.

Physically active persons with SCI differ from those who are inactive to a statistically significant degree, and with high effect size indexes in the area of emotional wellbeing. These results are supported by earlier research. This is reflected in an individual’s greater activity and energy [33]. Active individuals also showed a lower level of anxiety and depression and a higher level of extroversion [34], coupled with highly increased satisfaction and increased motivation for participating in PA [35]. A low level of anxiety and depression and high level of extroversion are also associated with a more positive acceptance of one’s new situation (self-acceptance, humor) and less frequent denial and avoidance, with positive emotions, greater self-confidence [36,37], and an inner locus of control over one’s life [37,38,39]. It is a major challenge for persons with SCI to believe that despite the significant changes they experience in all areas and despite all the problems they face after the injury, their life still makes sense. They maintain a sense of optimism, accept slow progress, experience satisfaction in life, accept an altered physical appearance, and experience feelings of safety and predictability. While the physical consequences of SCI are covered by very precise medical explanations and involve different well-targeted measures, it is more difficult to explain and understand the nature of invisible injuries that affect individuals in their broken inner worlds. Arguably, the emotional world of a person with SCI experiences a major breaking or turning point. Due to the loss of physical functions and the necessity to reorganize one’s life with a body that is no longer responding as it used to, tremendous power and ability are needed for the individual to adapt to new circumstances and start learning how to live anew. An individual with a “new” body, which is a source of instability and danger, can recover a sense of physical safety and emotional stability through the help and support of others, as well as with moderate, regular, and adjusted exercise. This can certainly take a long time, but only those who hope, fight, and persist can overcome the obstacles they face, and achieve greater emotional wellbeing.

The current study finds that physically active individuals manage their new lifestyle more independently. They feel that they have enough opportunities for independent choices (what they will do, where they will go, with whom they will socialize, etc.), while those who are inactive feel they have little or no choice for self-determination about their life and work. The results of past research are similar. The positive effects that engaging in PA after SCI have on a person’s physical wellbeing help them to develop more rapidly, especially in terms of increasing their remaining psychomotor abilities and gaining in strength [7,40]. This can increase functional independence, a sense of autonomy, and a sense of self-worth and efficacy, as well as the feeling of personal control over one’s life [11]. Having a passive attitude to the consequences of SCI (mental and behavioral passivity, avoidance of reality, a feeling of helplessness) and, above all, experiencing an external locus of control (a feeling of helplessness and the belief that others mostly decide on what happens to one’s own life) are associated with less satisfaction with life, lower psychological wellbeing, and, in turn, less opportunities to positively cope with the injury [37]. Despair, limited control over one’s own life, work, and choices, along with less sense of personal control and independence, are related to the negative perception of one’s own abilities and weaker social functioning [41,42]. Such negative emotional states with regard to dealing with the disability also increase the risk of suicide [1]. Being respected in one’s decisions and being able to engage in independent decision-making are also very much related to positive interpersonal relationships within the family or community [43].

The results indicate that individuals who are physically active are statistically significantly more satisfied with social inclusion. They are more satisfied with the support and care that they receive in their family, and more often spend free time with their family. They report that their relationships with family members are considerably positive, trustful, and friendly, and they are also more satisfied with their contacts with friends, neighbors, and acquaintances, as compared to those not physically active. PA offers many opportunities for socializing [10], helps decrease social isolation [44], and encourages the spending of free time in quality ways that enable people to “escape” their everyday routines [45]. These encourage faster reintegration and resocialization, as well as contribute to better psychological adjustment [46]. Positive interpersonal relationships [47], the relationship with their partners [42,48], adequate care in the family [42,49], support [43], close relationships with the family, and a higher level of participation in family activities, along with participation in various associations, and voluntary work [50], are seen to strongly encourage higher life satisfaction, the possibility to re-adopt some previously lost social roles, and better social integration [10]. With a positive attitude and encouragement, the family plays a very important role in the return of an individual with SCI to the community, and in reshaping the relationships and roles that have been lost with their injury [51]. Moral, physical, and financial support from the family, as well as encouragement and stimulation to engage in different activities and forms of socialization, give these individuals a feeling of safety and certainty. All this makes their “steps” along their new path easier, and in turn contributes to their increased satisfaction and QL. With a more frequent inclusion in different PA within the family, they are offered more opportunities for personal development, acquisition of the sense of personal and social responsibility, and of other social requirements. In this way, they also have more opportunities to become active members of the community, where they can be supported by other people, and where they have the opportunity to broaden the spectrum of their social ties [52]. It is in these two variables, namely, active inclusion in activities within the family and satisfaction with the role in the community, that the largest differences have been seen in both groups of persons considered in the present study, with regards to social inclusion. This leads us to conclude that physically active individuals generally receive more encouragement and support from the family compared to those who are inactive. According to Chang et al. [48], the QL of persons with SCI is seen to be very strongly determined by their social inclusion. Their higher general satisfaction with life, their ability to re-adopt certain social roles, and better social integration are seen to be strongly influenced mainly by their active engagement in the community, their successful integration, and their return and inclusion in the environment. Stephens et al. [32] further link social integration with increased opportunities of persons with SCI to learn from each other, and to acquire important information (such as healthy habits, functional development after the injury, and similar). It was revealed that the speed of social reintegration is determined by the level of PA: the higher the latter, the faster the reintegration [53]. Physically active persons more often engage in free-time activities, spend time with their family and friends, and give higher estimates for activities in the family than individuals who are not physically active. This can entail less social isolation, that after the SCI more often occurs, due to decreased social activity. However, both groups show a very similar share in their low estimates of active inclusion in various activities in their place of residence. The same applies for their sense of personal importance in their place of residence. The low degree of PA in the place of residence may be related to the lack of opportunities or adapted activities and facilities for them, since our respondents mainly lived in rural areas, while more opportunities and adjustments are offered in larger towns and cities [1]. Limited inclusion can further entail the feeling of being less important in the place, where these individuals live. Moreover, all respondents gave similar estimates of general satisfaction with the help and support provided by different social assistance services (such as volunteers, care providers, and social assistance). With regard to satisfaction with accessibility, support, and help provided by health care services, the present study shows that this variable is among those, in which persons with SCI who are physically active differ from those who are not, revealing a statistically significant medium effect size. Although a more detailed review shows that the estimates of satisfaction are low in both groups, individuals who are physically active are more satisfied than the other group with regard to accessibility, support, and help supplied by health care services. Literature provides data about social obstacles in the environment in which persons with SCI live that affect their activity and the related satisfaction with life after the injury [54]. The inaccessibility of healthcare and other care services, and lack of modern support technology for persons with SCI hinder the functionality and independence in their everyday life, and this is significantly related to experiencing the low QL [1,31,42]. This is the fact to be paid attention to by all social support services who offer help to persons with SCI. Accessibility, timeliness and adequacy of support, and helping services decrease their burden of concern increases their wellbeing and adaptation, and contributes to their higher QL.

Individuals who are physically active give higher estimates of their rights than those who are physically inactive. They preserve their dignity and report greater satisfaction in respect of their rights to equality, nondiscrimination, and privacy, as well as experiencing less environmental barriers. It is very important for a person with SCI, which very deeply interferes with the individual’s life, work, and relationships, that they return to their home environment, are well-received and respected there, including the regard for their rights to privacy, equality, nondiscrimination, and dignity. It is important that they have physical access to outer and inner environments (their own house or apartment, institutions, bus, library, school, work, sport facilities), and that the environment in which they live is adequately adapted to their new and different needs. Social influences of their own immediate and wider environment are importantly connected to their active approach to life, and in turn with life satisfaction after the injury [54]. An encouraging environment, infused with the spirit of moral, physical, and financial support and help enables a more intense experience of equality, privacy, and dignity. This involves activating their abilities rather than pointing to disabilities, as well as foregrounding happiness, joy, and personal pleasures. We can assume that individuals in our study who were physically active had positive experiences with all of these, and also that PA is designed to encourage relaxation, satisfaction, and joy in interpersonal relationships while also enabling the feeling of acceptance and equality, and preserving privacy and dignity. Undoubtedly, the experience of one’s rights being respected can also be provided by redesigning the environment, e.g., removing the barriers, including architectonic ones, enabling accessibility and practical use (adaptation) of public and private facilities, as well as regulated and accessible transport services and parking spaces [1]. Individuals engaging in PA as well as those who are not active expressed little or no satisfaction with regard to the consideration and respect of their rights to timely processes (in judicial, social care and others). According to Gething et al. [41], respect of human rights helps persons with SCI to be more economically and physically independent, have access to sufficient and adequate information, and enables them full social participation. Therefore, to reduce the risk of declining QL in persons with SCI, this factor should also be taken seriously in preventive action and interventions, and competent authorities should be encouraged to act accordingly. Only timely and adequate services (social, rehabilitation, judicial, and others) can contribute to the QL and better adaptation of persons with SCI [1,31,42], and in turn to the decreased risk of the decline in the quality of their life.

## 5. Conclusions

The research results show that QL was higher in 31 thoracic-level SCI (Th6–Th12) individuals who were physically active than in 31 inactive individuals. It could be argued that 62 responders is a small number of participants, but in fact it is the population sample of SCI with this level of injury in Slovenia. In addition, other variables (such as employment, education, place of residence) were controlled for in order to compare a homogeneous sample. A reliable and validated instrument was developed, which has not been used in any other study so far. The instrument was completed individually, and respondents voluntarily participated in the study. The instrument contains variables that map social domains (interpersonal relationships, social integration, and individual rights).

Quality of life for people with SCI has become increasingly important to care providers and policymakers. The research findings can help those who prepare and administer various assistance programs to design successful rehabilitation interventions, evaluate programs and assistance services, and meet the needs of people with SCI. In the context of the present study, it can be confirmed that PA plays an important role in different areas of QL, so it is useful to maintain and improve PA programs in different organizations. Another possibility is to create inclusive programs where people with and without disabilities PA participate together (e.g., swimming, tennis). Recently, Rehabilitation Centre in Ljubljana hired a new expert in kinesiology who will be one of the first implementers of PA in SCI. Recently, a new “Low of Sport” was introduced in Slovenia, which also focuses on people with disabilities (with more emphasis on competences for working with people with disabilities). The results of the present study need to be explored among persons with SCI to encourage and motivate persons with SCI to participate even more frequently in physical activity programs and inspire others to do so.

## Figures and Tables

**Table 1 ijerph-18-09148-t001:** Descriptive statistics for the variables that define the material prosperity of persons with SCI.

Material Prosperity	Status	Mean	SD	*t*	*p*	d
Satisfaction with income	physically inactive	2.29	0.86	2.99	0.004	0.77
	physically active	3.00	1.00			
Ability to take care of oneself and one’s family	physically inactive	2.42	0.85	4.55	0.000	1.17
	physically active	3.55	1.09			
Obstacles for employment	physically inactive	2.39	1.09	1.38	0.173	0.36
	physically active	2.81	1.30			
Ability to work	physically inactive	2.13	1.12	3.32	0.002	0.86
	physically active	3.26	1.53			

SD—standard deviation; *t*—*t* test score; *p*—statistical significance; d—effect size (Cohen’s d).

**Table 2 ijerph-18-09148-t002:** Descriptive statistics for the variables that define the physical wellbeing of persons with SCI.

Physical Wellbeing	Status	Mean	SD	*t*	*p*	d
Satisfaction with health	physically inactive	2.81	0.79	5.20	<0.001	1.34
	physically active	3.77	0.67			
Obstacles to daily tasks	physically inactive	2.65	0.92	1.73	0.090	0.45
	physically active	3.06	1.000			
Capacity to perform daily tasks	physically inactive	2.65	0.88	6.00	<0.001	1.55
	physically active	3.74	0.51			
Leisure activities	physically inactive	2.48	0.96	7.30	<0.001	1.88
	physically active	4.06	0.73			

SD—standard deviation; t—t test score; *p*—statistical significance; d—effect size (Cohen’s d).

**Table 3 ijerph-18-09148-t003:** Descriptive statistics for the variables that define the personal development of persons with SCI.

Personal Development	Status	Mean	SD	*t*	*p*	d
Education	physically inactive	2.77	1.06	2.89	0.005	0.75
	physically active	3.52	0.96			
Obstacles—Education	physically inactive	2.77	1.02	2.21	0.031	0.57
	physically active	3.35	1.05			
Concentration	physically inactive	3.06	0.73	3.73	0.000	0.96
	physically active	3.81	0.83			
Overcoming obstacles	physically inactive	3.00	0.73	5.52	0.000	1.43
	physically active	4.10	0.83			
Personal competence	physically inactive	2.90	0.65	4.76	0.000	1.23
	physically active	3.81	0.83			
Success	physically inactive	3.16	0.93	3.42	0.001	0.88
	physically active	3.81	0.48			
Productivity	physically inactive	2.84	0.86	5.98	0.000	1.54
	physically active	3.97	0.61			

SD—standard deviation; *t*—*t* test score; *p*—statistical significance; d—effect size (Cohen’s d).

**Table 4 ijerph-18-09148-t004:** Descriptive statistics for the variables that define the emotional wellbeing of persons with SCI.

Emotional Wellbeing	Status	Mean	SD	*t*	*p*	d
Life satisfaction	physically inactive	3.39	0.80	3.82	0.000 *	0.99
	physically active	4.10	0.65			
Optimism, hope, acceptance of reality	physically inactive	2.90	0.75	5.38	0.000 *	1.39
	physically active	4.00	0.86			
Meaning of life	physically inactive	3.00	0.86	5.56	0.000 *	1.44
	physically active	4.19	0.83			
Accepting of oneself	physically inactive	3.13	1.02	3.70	0.000 *	0.96
	physically active	4.00	0.82			
Slow progress	physically inactive	2.90	0.70	4.57	0.000 *	1.18
	physically active	3.84	0.90			
Religiosity	physically inactive	2.39	1.38	0.18	0.859	0.05
	physically active	2.45	1.46			
Safety	physically inactive	2.97	0.95	3.40	0.001 *	0.88
	physically active	3.77	0.92			

SD—standard deviation; *t*—*t* test score; *p*—statistical significance; d—effect size (Cohen’s d); * significant after Benjamini–Hochberg correction.

**Table 5 ijerph-18-09148-t005:** Descriptive statistics for the variables that define the self-determination of persons with SCI.

Self-Determination	Status	Mean	SD	*t*	*p*	d
Autonomy	physically inactive	2.90	0.75	7.23	0.000 *	1.87
	physically active	4.26	0.73			
Personal control	physically inactive	2.68	0.79	5.99	0.000 *	1.55
	physically active	3.84	0.74			
Respecting personal decisions, values	physically inactive	2.68	0.65	5.18	0.000 *	1.34
	physically active	3.48	0.57			
Personal choices	physically inactive	2.97	0.88	5.21	0.000 *	1.35
	physically active	4.10	0.83			
Independent decision making	physically inactive	3.55	1.18	3.45	0.001 *	0.89
	physically active	4.42	0.77			

SD—standard deviation; *t*—*t* test score; *p*—statistical significance; d—effect size (Cohen’s d); * significant after Benjamini–Hochberg correction.

**Table 6 ijerph-18-09148-t006:** Descriptive statistics for the variables that define the interpersonal relationships of persons with SCI.

Interpersonal Relationships	Status	Mean	SD	*t*	*p*	d
Good relations with friends	Physically inactive	3.74	0.82	2.63	0.011 *	0.68
	Physically active	4.26	0.73			
Leisure time with friends	Physically inactive	2.97	0.80	6.23	0.000 *	1.61
	Physically active	4.06	0.57			
Getting along with one’s family	Physically inactive	3.48	0.96	2.80	0.007 *	0.72
	Physically active	4.13	0.85			
Leisure time with family	Physically inactive	3.61	1.05	2.92	0.005 *	0.75
	Physically active	4.26	0.63			
Care and support in the family	Physically inactive	3.55	0.93	3.01	0.004 *	0.78
	Physically active	4.23	0.85			

SD—standard deviation; *t*—*t* test score; *p*—statistical significance; d—effect size (Cohen’s d); * significant after Benjamini–Hochberg correction.

**Table 7 ijerph-18-09148-t007:** Descriptive statistics for the variables that define social inclusion of persons with SCI.

Social Inclusion	Status	Mean	SD	*t*	*p*	d
Activities in the family	Physically inactive	3.16	0.93	4.93	0.000 *	1.27
	Physically active	4.26	0.82			
Activity in one’s place of residence	Physically inactive	2.42	0.89	1.84	0.071 *	0.48
	Physically active	2.90	1.17			
Feeling of importance	Physically inactive	2.61	1.02	1.44	0.156 *	0.37
	Physically active	3.00	1.10			
Role in the community	Physically inactive	3.13	0.89	3.82	0.000 *	0.99
	Physically active	3.94	0.77			
Social help and support	Physically inactive	2.94	0.73	0.47	0.638	0.12
	Physically active	3.03	0.88			
Accessibility and support of health services	Physically inactive	2.94	0.68	2.80	0.007 *	0.72
	Physically active	3.48	0.85			

SD—standard deviation; *t*—*t* test score; *p*—statistical significance; d—effect size (Cohen’s d); * significant after Benjamini–Hochberg correction.

**Table 8 ijerph-18-09148-t008:** Descriptive statistics for the variables that define the rights of persons with SCI.

Rights	Status	Mean	SD	*t*	*p*	d
Equality	Physically inactive	2.87	0.76	3.32	0.002 *	0.86
	Physically active	3.52	0.77			
Privacy	Physically inactive	3.39	0.72	2.13	0.037 *	0.55
	Physically active	3.77	0.72			
Dignity	Physically inactive	2.90	0.83	4.55	0.000 *	1.17
	Physically active	3.77	0.67			
Environmental obstacles	Physically inactive	2.84	0.69	2.33	0.023 *	0.60
	Physically active	3.23	0.62			
Timeliness of processes	Physically inactive	2.29	1.01	1.87	0.067	0.48
	Physically active	2.74	0.89			

SD—standard deviation; *t*—*t* test score; *p*—statistical significance; d—effect size (Cohen’s d); * significant after Benjamini–Hochberg correction.

## Data Availability

The data presented in this study are available on request. Please contact: tjasa.filipcic@pef.uni-lj.si.

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
