# Peer review of "Quality of Life and Physical Activity of Persons with Spinal Cord Injury"

_ijerph, 2021, doi:10.3390/ijerph18179148_

Round 1
Reviewer 1 Report
I am sorry to say that I do not consider that your manuscript merits being published. Main reasons include:
- Background: The background does not provide a basis for the hypothesis or the objectives. Is too long but lack important concepts, such as the importance of activity for neurological recovery. The text includes multiple mistakes including a repeated paragraph as well as statements that are, at least, arguably or need to be rephrased. For example: "Persons with spinal cord injury (SCI) report good and even excellent quality of life [1], while outcomes compared to those without health problems can range from well below average [2] to above average [3]".
- Objetives: the written aim ("...to determine whether there statistically significant differences in subjectively perceived quality of life between physically active and inactive persons with SCI..."), justified on the fact that "there are no comparable studies comparing physically active and inactive individuals" is just a mistake. There are multiple studies comparing active and inactive SCI patients (see for example multiple studies cited in Tomasone et al. 2013. Kinesiology Review, 2(2), 113-129). Therefore, the background does not support the need for this study.
- Design: additional data is always a valuable resource that merits publication in my opinion. However, the design is too simple. For example, you have not consider the degree of impairment of the patients and it may be related to their degree of activity and also explain many of the QoL variables.
- Methods: I would like to see the questionnaire, to know how do you classify patients between active and inactive, and many the details. Are the data deposited somewhere? There are many mistakes or confusing texts such as the inclusion criterium ".... age at time of measurement between 18 and 50 years" which contradicts the 55.2 years.mean age of inactive males.
- Results: a list of tables comparing active and inactive values results poor. Also, there is no multiple testing correction required by the present analysis (vg. Benjamini and Hochberg FDR). A multivariate analysis may provide interesting associations between variables. Anyway, improved analysis cannot solve the initial design limitations.
In summary, in my opinion, the manuscript and the whole study needs further work before going for publication.
Author Response
Dear Reviewer.
We are very thankful for your very constructive and challenging comments, and especially for the excellent suggestions regarding methodology and objectives. We tried to address your suggestions to our best ability. We believe your thorough reading, in-depth understanding and excellent suggestions helped us to substantially improve the quality of this manuscript, and that the revised manuscript is suitable for publication. Our answers to your comments are in red and MS word function “track changes” in manuscript is enabled.
Reply to the 1st reviewer
1st comment:
- Background: The background does not provide a basis for the hypothesis or the objectives. Is too long but lack important concepts, such as the importance of activity for neurological recovery. The text includes multiple mistakes including a repeated paragraph as well as statements that are, at least, arguably or need to be rephrased. For example: "Persons with spinal cord injury (SCI) report good and even excellent quality of life [1], while outcomes compared to those without health problems can range from well below average [2] to above average [3]".
Answer to the 1st reviewer
Thank you very much for your comment. We have made a complete revision of Introduction where we introduced basis for the objectives. The background is shorter and after examined multiple studies of QL and PA we found a support to justify our study. We focused on three areas that also followed suggestions of Martin Ginis (2010) and Tomasone et al. (2013). Therefore, the aim of our study was to investigate the subjective QL, considering also the social domain what was somehow lack in previous research. We developed instrument to measure QL among people with SCI (in previous studies instruments were usually take from other research and research did not show metric characteristics. Finally, we emphasized homogeneity of the sample (Th 6-Th 12 SCI).
We kindly ask you to review this part of the article again.
2nd comment:
Objectives: the written aim ("...to determine whether there statistically significant differences in subjectively perceived quality of life between physically active and inactive persons with SCI..."), justified on the fact that "there are no comparable studies comparing physically active and inactive individuals" is just a mistake. There are multiple studies comparing active and inactive SCI patients (see for example multiple studies cited in Tomasone et al. 2013. Kinesiology Review, 2(2), 113-129). Therefore, the background does not support the need for this study.
Answer to the 1st reviewer
Thank you very much for your comments and suggestions. After the background was rewritten it was easier to find objectives of the present study. To follow the suggestions of Martin Ginis (2010) and Tomasone et al. (2013), the aim of our study was to investigate the subjective QL, considering also the social domain. We developed instrument to measure QL among people with SCI. The focus was whether there are statistically significant differences in subjectively perceived QL between physically active and inactive individuals with SCI (Th 6-Th 12) in terms of material well-being, physical and emotional well-being, personal development, self-determination, interpersonal relationships, social integration and rights.
3rd comment:
Design: additional data is always a valuable resource that merits publication in my opinion. However, the design is too simple. For example, you have not consider the degree of impairment of the patients and it may be related to their degree of activity and also explain many of the QoL variables.
Answer to the 1st reviewer
Thank you very much for your comment. We added additional data. As far as degree of impairment this variable was also included after your comment. Namely, all 62 participants had injury between Th 6 -Th 12 level). No statistical differences were found among group as far as level of injury was concerned. In addition, other variables (such as marital status, employment, education, place of residence) were controlled. No statistical differences were found among groups.
4th comment:
Methods: I would like to see the questionnaire, to know how do you classify patients between active and inactive, and many the details. Are the data deposited somewhere?
Answer to the 1st reviewer
Thank you very much for your willingness. The questionnaire is in Slovene language but we translated it into English. We classified responders as physically active/inactive according to WHO guidelines on physical activity and sedentary behaviour (2020).
Respondents were asked to define themselves as active when they reached at least 150 minutes of moderate intensity aerobic physical activity or at least 75 minutes of vigorous intensity aerobic physical activity throughout the week) or inactive (not engage in any physical activity that would reach criteria for active responders). That was also added in the text.
We will attach the questionnaire to show how QL was assessed.
We have all results in SPSS programme and can be sent to you if needed.
5th comment:
There are many mistakes or confusing texts such as the inclusion criterium ".... age at time of measurement between 18 and 50 years" which contradicts the 55.2 years.mean age of inactive males”.
Answer to the 1st reviewer: “
Thank you very much for your comment. We have calculated all data again (mean age and SD of participants) and corrected mistakes.
6th comment:
Results: a list of tables comparing active and inactive values results poor. Also, there is no multiple testing correction required by the present analysis (vg. Benjamini and Hochberg FDR). A multivariate analysis may provide interesting associations between variables. Anyway, improved analysis cannot solve the initial design limitations.
Answer to the 1st reviewer: “
Thank you very much for your comment. To account for multiple testing, we adjusted p values of these associations with the use of the Benjamini-Hochberg false discovery rate procedure (FDR).
7th comment
In summary, in my opinion, the manuscript and the whole study needs further work before going for publication.
Thank you very much for your time and willingness to improve our paper. We were trying to consider most of your comments and suggestions. We kindly ask you to review our paper again.
In additional text you will find our Instrument/questionnaire.
VPRAŠALNIK/QUESTIONNAIRE
Questionnaire is divided in 3 parts: demographic data, PA and Quality of Life among persons with SCI
All translation in English is in red colour. In black color we kept Slovene language.
- Splošni podatki/DEMOGRAFIC DATA
Spol/GENDER: a) moški/MEN b) ženski/FEMALE
Starost/AGE______
Starost v času poškodbe/AGE WHEN INJURY OCCURED _____
Čas pretečen od poškodbe/YEARS SINCE THE INJURY OCCURRED _____
Nivo poškodbe/LEVEL OF INJURY: _____________________________
Popolna/nepopolna COMPLETE/INCOMPLET SCI
MARITUAL STATUS:
- a) poročen/v partnerski zvezi/MARRIED
- b) ločen/DIVORCE
- c) vdovec/WIDOW
- d) samski/SINGLE
Kraj bivanja/PLACE OF RESIDENCE
- a) velemesto/LARGE CITY
- b) manjše mesto/SMALL TOWN
- c) vas/VILLAGE
Ali je bivališče (hiša/stanovanje)/ADAPTATION OF RESIDANCE
- Prilagojeno/ ADAPTED
- Ne prilagojeno/ NOT ADAPTED
Izobrazba/EDUCATION
- a) osnovna šola/PRIMARY SCHOOL
- b) srednja šola/SECONDARY SCHOOL
- c) višja/ HIGH SCHOOL
- e) univerzitetna/UNIVERSITY
- f) magisterij/MASTER OF SCIENCE
- g) doktorat/PhD
Zaposlitev/EMPLOYMENT
- zaposlen za polni delovni čas/FULL TIME
- zaposlen za krajši delovni čas/HALF TIME
- nezaposlen/UNEMPLOYED
- upokojen/RETIRED DUE TO SCI INJURY
- študent/STUDENT
- SUBJEKTIVNA OCENA TELESNIH DEJAVNOSTI/
SUBJECTIVE ESTIMATION OF PHYSICAL ACTIVITY IN LAST MONTH
Telesna dejavnost/PHYSICAL ACTIVITY
Pomislite na zadnjih 14 dni in ocenite vašo gibalno dejavnost/ Estimate your physical activity in last month.
- ESTIMATE YOUR PHYSICAL ACTIVITY (CHOOSE ONE)
1: physically active: I have reached at least 150 minutes of moderate intensity aerobic physical activity or at least 75 minutes of vigorous intensity aerobic physical activity throughout the week)
2: physically inactive (not engage in any physical activity that would last at least 150 minutes of moderate intensity aerobic physical activity or at least 75 minutes of vigorous intensity aerobic physical activity throughout the week
- Če ste športno dejavni, izpolnite spodnjo tabelo, če niste, pojdite na naslednje vprašanje. V tabeli z X označite, s katerimi aktivnostmi se ukvarjate, ter način ukvarjanja: R-rekreativno, T-tekmovalno/IF YOU ARE PHYSICALLY ACTIVE SIGN YOUR PARTICIPATION IN SPORT AND LEVEL OF YOUR PARTICIPATION (WITH X).
|
SPORT |
RECREATION LEVEL |
COMPETING LEVEL |
|
Atletika/ATHLETICS |
|
|
|
Košarka/BASKETBALL |
|
|
|
Plavanje/SWIMMING |
|
|
|
Namizni tenis /TABLE TENNIS |
|
|
|
Tenis/TENNIS |
|
|
|
Keglanje/BOWLING |
|
|
|
Alpsko smučanje /ALPINE SKIING |
|
|
|
Kolesarjenje/CYCLING |
|
|
|
Lokostrelstvo/ARCHERY |
|
|
|
Strelstvo/SHOOTING |
|
|
|
Ples/DANCE |
|
|
|
Ostalo/OTHER: |
|
|
- SUBJEKTIVNA OCENA KAKOVOSTI ŽIVLJENJA
SUBJECTIVE QL
Prosimo vas, da pomislite na svoje življenje v zadnjem mesecu. Imejte v mislih svoje vrednote, upanja, radosti in skrbi. Če ste negotovi, je prvi odziv najpogosteje najboljši. S številkami od 1 do 5 ocenite/ Please define:
|
ALI STE ZADOVOLJNI /SATISFACTION WITH… |
Very unsatisfied |
Not satisfied
|
Little satisfaction
|
satisfied |
Very satisfied |
|
S KOLIČINO DENARJA, KI GA DOBITE? Satisfaction with your income |
1 |
2 |
3 |
4 |
5 |
|
Z ZDRAVJEM/ZDRAVSTVENIM STANJEM? Satisfaction with your health status |
1 |
2 |
3 |
4 |
5 |
|
S KAPACITETAMI (fizičnimi, psihičnimi) ZA OPRAVLJANJE VSAKODNEVNIH DEJAVNOSTI? Satisfaction with capacities to perform daily tasks |
1 |
2 |
3 |
4 |
5 |
|
Z USPEHI V ŽIVLJENJU (na različnih področjih)? Satisfaction with experiencing success (accomplishments and achievements in life) |
1 |
2 |
3 |
4 |
5 |
|
S STIKI, KI JIH IMATE S PRIJATELJI, SOSEDI, ZNANCI (so sproščeni, pozitivni)? Satisfaction with interactions with friends, roommates, neighbours, and acquaintances |
1 |
2 |
3 |
4 |
5 |
|
Z OSKRBO IN PODPORO V DRUŽINI? Satisfaction with care and support in the family |
1 |
2 |
3 |
4 |
5 |
|
Z VLOGO V SKUPNOSTI (kot sodelavec …) Satisfaction with role in the community (as co-worker, volunteer) |
1 |
2 |
3 |
4 |
5 |
|
Z DOSTOPNOSTJO, PODPORO IN POMOČJO ZDRAVSTVENIH STORITEV? Satisfaction with access to, support and assistance of health service |
1 |
2 |
3 |
4 |
5 |
|
S SPOŠTOVANJEM IN UPOŠTEVANJEM PRAVIC DO ENAKOSTI IN NEDISKRIMINACIJE? Satisfaction with experiencing and enforcing the rights to equality and non-discrimination |
1 |
2 |
3 |
4 |
5 |
|
Z UPOŠTEVANJEM PRAVIC DO ZASEBNOSTI Satisfaction with the possibility of privacy, retreat |
1 |
2 |
3 |
4 |
5 |
|
Z ZMANJŠEVANJEM OVIR V VAŠI OKOLICI? Satisfaction with accessibility, removal of environmental barriers |
1 |
2 |
3 |
4 |
5 |
|
Z UPOŠTEVANJEM IN SPOŠTOVANJEM PRAVIC DO PRAVOČASNOSTI (sodnih, socialnih idr.) PROCESOV? Satisfaction with justice and timeliness in judicial, medical and other processes |
1 |
2 |
3 |
4 |
5 |
|
ESTIMATE YOUR INDIVIDUAL ABILITY/OBSTACLES/EXPERIENCES |
Not at all |
Little |
Considerably |
A lot
|
Entirely
|
|
LAHKO FINANČNO SKRBITE ZASE/ZA SVOJO DRUŽINO? Do you have ability to take care of yourself and your family? |
1 |
2 |
3 |
4 |
5 |
|
VAS POSLEDICE POŠKODBE OVIRAJO PRI OPRAVLJANJU DELA OZIROMA ISKANJU ZAPOSLITVE? Do you have obstacles to work or to find employment due to the secondary conditions (SCI) |
5 |
4 |
3 |
2 |
1 |
|
VAS POSLEDICE POŠKODBE OVIRAJO PRI DNEVNIH OPRAVILIH? Do you experience obstacles to daily tasks due to secondary conditions (SCI) |
5 |
4 |
3 |
2 |
1 |
|
MOČNA JE V VAS ŽELJA PO UČENJU NOVEGA? How strong is your desire for education (learn something new) |
1 |
2 |
3 |
4 |
5 |
|
VAS POSLEDICE POŠKODBE OVIRAJO PRI USVAJANJU NOVEGA ZNANJA IN VEŠČIN? Do you experience obstacles to assimilate new knowledge due to secondary conditions (SCI) |
5 |
4 |
3 |
2 |
1 |
|
SE LAHKO SKONCENTRIRATE? Do you experience ability to focus on work (can you concentrate on learning and work |
1 |
2 |
3 |
4 |
5 |
|
ZMORETE SAMOSTOJNO UPRAVLJATI NOV NAČIN ŽIVLJENJA S POŠKODBO? Do you experience the ability to independently manage a new way of life |
1 |
2 |
3 |
4 |
5 |
|
IMATE OBČUTEK, DA SO UPOŠTEVANE VAŠE ODLOČITVE, VREDNOTE, MNENJA, ŽELJE IN PRIČAKOVANJA? Do you experience a sense of personal control and autonomy (independence) |
1 |
2 |
3 |
4 |
5 |
|
ČUTITE, DA LAHKO SAMOSTOJNO IZBIRATE (kaj boste počeli, kam boste šli, s kom se boste družili, koliko denarja boste porabili …)? Do you have possibility of independent choices (where to go, with whom, how to spend money, etc.) |
1 |
2 |
3 |
4 |
5 |
|
KOLIKO/DO YOU EXPERIENCE: |
Not at all |
Little |
Considerably |
A lot
|
Entirely
|
|
DRUGI ODLOČAJO O VAŠEM ŽIVLJENJU, DELU Do you experience others making decision about your life/work |
5 |
4 |
3 |
2 |
1 |
|
SO VAŠI ODNOSI Z DRUŽINSKIMI ČLANI POZITIVNI, ZAUPLJIVI, ODPUŠČAJOČI, PRIJATELJSKI, NEKONFLIKTNI? Do you experience compassionate, intimate, friendly relationships with family members |
1 |
2 |
3 |
4 |
5 |
|
STE AKTIVNO VKLJUČENI V DEJAVNOSTI V OKVIRU SVOJE DRUŽINE (izleti, druženja …)? Do you experience leisure time with your family? |
1 |
2 |
3 |
4 |
5 |
|
SE AKTIVNO VKLJUČUJETE V DEJAVNOSTI V KRAJU, V KATEREM ŽIVITE? Are you actively in your place of residence? |
1 |
2 |
3 |
4 |
5 |
|
DOŽIVLJATE, DA STE POMEMBEN ČLAN KRAJA, V KATEREM ŽIVITE? Do you experience a sense of importance in your community? |
1 |
2 |
3 |
4 |
5 |
|
LAHKO PRIČAKUJETE POMOČ IN PODPORO DRUŽBENIH PODPORNIH SLUŽB (prostovoljci, servisi za pomoč, asistenco, socialna služba … )? Do you experience potentional for social support (support networks, assistance) |
1 |
2 |
3 |
4 |
5 |
|
DOŽIVLJATE, DA LAHKO OHRANJATE DOSTOJANSTVO? Do you experience dignity? |
1 |
2 |
3 |
4 |
5 |
|
ALI/DO YOU?: |
Not at all |
Little |
Considerably |
A lot
|
Entirely
|
|
SE POČUTITE DOVOLJ SPOSOBNI ZA OPRAVLJANJE DELA V SLUŽBI (če jo imate ali bi jo lahko imeli)? Do you have sense of ability to work? |
1 |
2 |
3 |
4 |
5 |
|
VERJAMETE VASE, DA ZMORETE USPEŠNO PREMAGOVATI OVIRE IN SPREJETI TUDI NEUSPEH? Do you feel you can overcome obstacles an also except failure? |
1 |
2 |
3 |
4 |
5 |
|
STE NA SPLOŠNO ZADOVOLJNI S SVOJO USPOSOBLJENOSTJO ZA ŽIVLJENJE, UČENJE, ODNOSE? Are you satisfied with your personal competencies (cognitive, social, practical) |
1 |
2 |
3 |
4 |
5 |
|
GOJITE OPTIMIZEM, UPANJE, POZITIVNO IN REALNO SLIKO O PRIHODNOSTI? Do you have positive acceptance of life? |
1 |
2 |
3 |
4 |
5 |
|
JE VAŠE ŽIVLJENJE VREDNO IN SMISELNO? Is your life worth living and meaningful? |
1 |
2 |
3 |
4 |
5 |
|
ZMORETE SPREJETI TELESNI VIDEZ IN SEBE KOT EDINSTVENO OSEBNOST? Can you accept yourself (your identity, self-esteem). |
1 |
2 |
3 |
4 |
5 |
|
ZMORETE SPREJEMATI DEJSTVO, DA NAPREDUJETE POČASI (ste potrpežljivi sami s seboj)? Can you accept slow progress? Are you patient? |
1 |
2 |
3 |
4 |
5 |
|
VAM K VESELJU DO ŽIVLJENJA PRIPOMORE VERA, ZAUPANJE V BOGA? Can religion help you to find peace and certainty ? |
1 |
2 |
3 |
4 |
5 |
|
SE ČUTITE VARNE? Do you feel secure? |
1 |
2 |
3 |
4 |
5 |
|
MENITE, DA JE VAŠE ŽIVLJENJE KAKOVOSTNO? Do you experience satisfaction in life (feelings of enjoyment and a good mood), stress reduction (feelings of predictability and control) |
1 |
2 |
3 |
4 |
5 |
|
KAKO POGOSTO: How often |
Never
|
Rarely |
Occasionaly |
Very often |
Always |
|
SI VZAMETE ČAS ZA SPROSTITEV, PROSTOČASNE DEJAVNOSTI (hobije, rekreacijo)? Do you take time for leisure activities and hobbies)? |
1 |
2 |
3 |
4 |
5 |
|
DOŽIVLJATE, DA STE USPEŠNI IN PRODUKTIVNI V TEM, KAR POČNETE? Do you experience productivity in your life? |
1 |
2 |
3 |
4 |
5 |
|
PREŽIVLJATE PROSTI ČAS S PRIJATELJI, SOSEDI, ZNANCI, SOIGRALCI …? Do you spend time with friends, neighbours, and others? |
1 |
2 |
3 |
4 |
5 |
|
PREŽIVLJATE ČAS Z DRUŽINO? Do you spend time with your family? |
1 |
2 |
3 |
4 |
5 |
Reviewer 2 Report
In this report the authors inquired the quality of life
of persons with spinal injury in a group of Slovenia
patients. They found that involvement with sports
sufficiently improved the patients' interpersonal relations
and general satisfaction in quality of living.
Author Response
We are very thankful for your in-depth reading and we thank you for positive feedback.
Authors
Reviewer 3 Report
This is a study of people with spinal cord injury (SCI).
The aim was to establish whether individuals with SCI who are active in sport subjectively rate their quality of life as higher compared to those who do not do sport.
Sixty-two respondents from Slovenia with spinal cord injury were interviewed. Thirty-one of them were involved in sport and some were not.
I believe that the results of this study have a high external validity and may encourage people with spinal cord injury to participate more often in sports programmes and also to encourage others to do so.
The study is very simple methodologically and perhaps the sample is very small. But I believe that science is there to help improve people's quality of life and I think that the results obtained in this study could help many professionals to encourage people with spinal cord injury to practice sport and of course people with spinal cord injury to take up physical activity. Overall I think it is a great article.
Summary: Essentially the most relevant part of the article.
Introduction: Very complete and very clarifying of the subject to be dealt with.
Methodology:
The authors should express what the total potential sample was. And why have they obtained only 62 subjects?
How and by whom were the questionnaires sent and, above all, how many did not respond or were filled in with errors or incomplete?
How much time did they have to fill in the questionnaire, was it face-to-face or online (or both possibilities)?
Results: Analysis well done
Discussion: I think they should add a small paragraph with suggestions to attract and/or motivate other spinal cord injured people to practice sport.
Author Response
Dear reviewer.
We are very thankful for your very constructive and challenging comments, and especially for the excellent suggestions regarding methodology. We tried to address your suggestions to our best ability. We believe your thorough reading, in-depth understanding and excellent suggestions helped us to substantially improve the quality of this manuscript, and that the revised manuscript is suitable for publication. Our answers to your comments are in red and MS word function “track changes” in manuscript is enabled.
Our answers
1st Comment
The study is very simple methodologically and perhaps the sample is very small.
Response:
Thank you very much for this comment. In Slovenia we have 1076 members with acquired or congenital spinal cord disability. 43% of 1076 are paraplegics with complete SCI. Therefore, our sample of 62 individuals represents 13.4% of the selected population. Moreover, a homogeneous sample was included. Namely, all participants were injured in the thoracic part of the spine (Th 6-Th 12), which would represent even larger percentage of the Slovene population. Additional data on the sample were added and described in the Methods chapter. Information regarding Level of injury was limitations of previous studies where researchers included more participants but the sample was heterogeneous (e.g., 34% paraplegics and 66% quadriplegics). Regarding the methodology, we must mention that we used our instrument to study the domains of QL and all the necessary statistical methods were considered.
2nd Comment
Regarding the methodology: the authors should indicate how large the total potential sample was. And why did they only get 62 subjects?
Response:
Thank you for your question. The potential sample would be 463, but since participation in the study was voluntary, therefore only 62 subjects participated. This represents the representative sample of Slovenian individuals with SCI (level of injury from Th 6-Th12).
3rd Comment
How and by whom were the questionnaires sent and, most importantly, how many did not respond or were filled in incorrectly or incompletely? How much time did they have to complete the questionnaire, was it in person or online (or both options)?
Response:
Thank you for your question. The survey was first presented to the regional organisation SCI and its members with SCI. 62 responded and agreed to participate. Later, personal contact was made with individual participant with SCI, so this was a face-to-face contact. Each individual had sufficient time to complete the questionnaire and received additional clarification. Therefore, all 62 questionnaires were considered and analysed. We have included this additional text where the research design is presented.
4th Comment
Discussion:I think a small paragraph should be added with suggestions to attract and/or motivate other spinal cord injury patients to participate in sports.
Response:
Thank you very much for your comment. A small paragraph has been added in conclusion where we put emphasis on policy makers, experts and also on individuals with SCI.
Finally, thank you for your time and willingness to improve our work. We have tried to address most of your comments and suggestions.